# Antimicrobial susceptibility profile of Gonococcal isolates obtained from men presenting with urethral discharge in Addis Ababa, Ethiopia: Implications for national syndromic treatment guideline

**Surafel Fentaw**[1]*, **Rajiha Abubeker**[1], **Negga Asamene**[1], **Meseret Assefa**[1], **Yonas Bekele**[2], **Eyasu Tigabu**[1,3]

1 Clinical Bacteriology and Mycology, Ethiopian Public Health Institute Ethiopia, Addis Ababa, Ethiopia, 2 Center for Cancer Research, National Cancer Institute, National Institute of Health, Bethesda, MD, United States of America, 3 Global One Health Initiative, The Ohio State University, Columbus, OH, United States of America

* sura4f@gmail.com

## Abstract

### Background

*Neisseria gonorrhoeae* (gonococcus) is the etiologic agent for the sexually transmitted Infection gonorrhea, a disease with a significant global public health impact. The treatment regimen for gonorrhea has been changed frequently over the past few decades due to the organism's propensity for developing antibiotic resistance. This study investigated antimicrobial susceptibility patterns of quinolones, third-generation cephalosporin, and other relevant antimicrobials found in *N. gonorrhoeae* isolated from men presenting with urethral discharge at selected healthcare facilities in Addis Ababa, Ethiopia, with the aim of revising the national treatment regimen based on the information generated from this study.

### Methods

A total of 599 male patients presenting with urethral discharge were included in the current study. Urethral discharge specimens were cultured on Modified Thayer Martín media and suspected *gonococcal* colonies were confirmed using Oxidase and Superoxol tests followed by identification through a commercial kit (API-NH^R). Antimicrobial susceptibility testing was performed by the Kirby-Bauer disc diffusion method using ciprofloxacin (5μg), ceftriaxone (30μg), cefixime (5μg), cefoxitin (30 μg), penicillin (10μg) and spectinomycin (100 μg) on enriched GC agar. Minimum Inhibitory Concentration (MIC) was also carried out using concentration gradient strips (E-tests) of the same antimicrobial agents.

### Results

The prevalence of gonococcal isolates in the current study was 69%. Out of the 361 gonococcal isolates, close to 68% were fluoroquinolone non-susceptible, with 60% resistant and

**Data Availability Statement:** All relevant data are included in the paper and its Supporting Information files. Additional raw data are available upon request to the corresponding author.

**Funding:** The fund was from CDC Atlanta through PEPFAR 1045 Cooperation agreement to Ethiopia. Specifically, the fund for STI ethological study to SF. The funder had no role in study design, data collection and analysis, decision to publish, or preparation of the manuscript.

**Competing interests:** The authors have declared that no competing interests exist.

7% having an intermediate status. However, all tested isolates were susceptible to ceftriaxone. In addition, all of the isolates have shown reduced non-susceptibility to spectinomycin and cefoxitin.

## Conclusion

The prevalence of gonococcal isolates in men presenting with urethral discharge at selected healthcare facilities in Addis Ababa, Ethiopia was found to be high. The high level of fluoroquinolone resistance observed in gonococcal isolates recovered in this study necessitates revision of the national syndromic treatment guideline.

## Background

*Neisseria gonorrhoeae* (gonococcus) is the second most common etiologic agent known to cause sexually transmitted infection (STI) with a significant global public health impact [1]. In uncomplicated infections, the disease manifests as urethritis in men and mucopurulent cervicitis in women. Because gonococcal infections are often asymptomatic in women, the lack of noticeable symptoms may result in serious complications such as pelvic inflammatory disease, ectopic pregnancy, and infertility. Infants born to a mother with the infection could develop conjunctivitis which may eventually lead to blindness if the infection is left untreated. In men, if left untreated, the disease could result in problems such as epididymitis, urethral stricture, and infertility. The transmission of the disease occurs by direct contact with secretions of infected mucosal surfaces and the incubation period can range from 1 to 10 days [1]. Gonorrhea, as with any STI, can work as a gateway to HIV and other infections [2, 3]. Globally, more than millions of people are affected by curable STIs. According to WHO, in 2012 alone, globally, there were an estimated 78 million new case of gonococcal diseases [1]. Public health control of gonorrhea requires both treatment of patients with appropriate antimicrobials as well as generalized and targeted prevention efforts [1, 4, 5].

Treatment regimens for gonorrhea have been changed frequently over the past few decades due to the organism's propensity for developing antibiotic resistance [1]. Over the past few years, gonococcus have become less susceptible to previously used antibiotics such as ciprofloxacin or tetracycline [1]. Until recently, quinolones have been used as an alternative to treat gonococcal infections. However, the emergence and spread of gonococci resistant to the quinolone group and reduced susceptibility to third generation cephalosporin antibacterial was reported from different corners of the world [1]. This trend is concerning considering no alternative antibiotic treatment options or combinations have been proven to be effective against the organism [6–10].

The treatment for gonococcal infection in sub-Saharan Africa countries, including Ethiopia, is based on a syndromic approach using single dose fluoroquinolone treatment. The basis for this regimen was under the assumption that resistance to fluoroquinolones is considered to be low in Africa. However, with the occurrence of resistance to commonly prescribed antibiotics in both developed and developing countries, it is imperative to investigate the resistance pattern of gonococcal isolates periodically. Therefore, updated knowledge of the prevailing susceptibility patterns of gonococcal isolates in Ethiopia is important for the proper selection and use of antimicrobial drugs as well as for the development of an appropriate prescription policy. Therefore, this study aimed to investigate the susceptibility patterns of quinolones and

third-generation cephalosporin found in *N. gonorrhoeae* isolated from urethral discharge of male patients seen in selected Addis Ababa city health centers.

## Methods

### Study sites and design

This study was conducted within the Addis Ababa, Ethiopia City Administration. Addis Ababa, the capital city of the Democratic Republic of Ethiopia, is geographically located in the central part of the country. A cross-sectional, facility-based study was conducted in eight healthcare centers of Addis Ababa. The selected healthcare facilities consisted of the *Arada*, *Tekalehaimanot*, *Addis-Ketema*, *Kirkos*, *Kotebe*, *Akaki-Kaliti*, *Shiromeda*, and *Kassanchis* health centers. These healthcare facilities were selected based on a high flow of STI patients determined from a previous assessment. The study team collected samples from visiting patients over a span of twelve months at each study site following training on study protocols, procedures, and research ethics.

### Source population and study participants

The source population consisted of patients visiting the selected healthcare facilities within Addis Ababa, Ethiopia with symptoms of urethral discharge who also gave consent to participate in the study. All urethral discharge specimens analyzed between August 2013 and August 2014 were included in this study.

### Laboratory methods

**Specimen collection.** Men presenting to the selected healthcare facilities with urethral discharge syndrome were recruited in the study following their consent. Afterward, a sterile Dacron swab-tipped applicator was used to collect urethral secretions. The swabs were then inoculated on Modified Thayer Martin Agar plates made of Gonococcal agar base supplemented with isovitalex (vitox); vancomycin, colistin, nystatin, and trimethoprim (VCNT); and synthetic hemoglobin (Oxoid and BBL) prepared in-house. The inoculated plates were incubated on site using a candle jar and then transported to the Ethiopian Public Health Institute (EPHI), Clinical Bacteriology and Mycology Reference Laboratory within the same day of collection. Swabs were rolled onto a microscopy slide, labeled, heat fixed, placed in a slide box, and sent to EPHI for Gram-stain analysis.

**Culture and identification.** In the clinical bacteriology laboratory at EPHI, inoculated plates were incubated at 35˚C in a carbon dioxide enriched environment (5–8% $CO_2$) for 72-hours. Plates were inspected every day for the growth of small, translucent, and non-pigmented colonies. Plates that were gram-negative, diplococcic, convex, glistening, elevated, had mucoid colony characteristics, and were oxidase, catalase, and supercool (30% $H_2O_2$) positive were considered as probable *N. gonorrhoeae* and further confirmed by carbohydrate and enzymatic tests using API-NH[R]. Antimicrobial susceptibility testing was performed by the Kirby-Bauer disc diffusion method using ciprofloxacin (5 μg), ceftriaxone (30 μg), cefixime (30 μg), cefoxitin (30 μg), penicillin (10 μg) and spectinomycin (100 μg) on enriched GC agar (Oxoid Ltd) plus 1% BBL Isovitalex Enrichment. Minimum Inhibitory Concentration (MIC) was done using concentration gradient strips (E-test) of the same antibiotics. The range of inhibition zones and MIC for each type of antibiotic disk were interpreted according to Clinical Laboratory Standard Institute (CLSI) guidelines [11]. *Neisseria gonorrhoeae* reference strain ATCC 49226 was used as a positive control.

## Data extraction methods

A structured checklist was used to collect information on socio-demographics, clinical history, sexual behaviors, pro-antibiotics taken, and laboratory data such as the antibiotic susceptibility results. All data were double entered to Cespro 8 software by two individuals and data analysis was done using SPSS version 20.

## Operational definitions

Non-susceptible *N. gonorrhoeae* isolates were defined as those that are not sensitive to the antibiotic tested for susceptibility, i.e., those isolates exhibiting resistance or intermediate resistance.

Dual non-susceptibility was defined as lack of susceptibility to any two of the antibiotics tested for susceptibility. Multi Drug Non-susceptibility was defined as combined non-susceptibility to an injectable cephalosporin and any two of either quinolones, penicillins, or tetracyclines.

## Ethics and consent to participate

This study was ethically cleared by the Scientific and Ethical Review Office (SERO) of the Ethiopian Public Health Institute and the Institutional Review Board (IRB) of CDC-Atlanta. At the enrollment visit, all men with urethral discharge provided written consent after being diagnosed according to the syndromic treatment guidelines approved in Ethiopia. Those who were eligible ($>$ 18 years of age) and willing to participate in the study were recruited using a structured questionnaire for their demographic and behavioral data. All data were kept confidential and anonymous. Brief counseling on the importance of adherence to STI medications, not having sex while taking medications, HIV/STI prevention, and recommendations to use condoms to reduce STI/HIV acquisition and transmission was also given.

# Results and discussion

Between August 2013 and August 2014, a total of 599 urethral discharge specimens were collected from male patients visiting one of the eight selected healthcare centers for routine clinical care and the collected specimens were microbiologically analyzed. The mean age of the study participants was 27 years (SD ± 7.2), with all being male. Observation of the urethral discharge specimens revealed that over 90% of them were profuse/thick discharge (Table 1).

## Proportion of gonococcal isolates recovered

Of all the specimens analyzed, 415 (69.3%) gonococcal isolates were identified through culture methods. Compared to culture, the proportion of presumptive gonorrhea-positive samples

**Table 1. Clinical presentations of urethral discharge from patients visiting health centers in Addis Ababa.**

| Clinical feature | Category | N (%) |
|---|---|---|
| **Fluid coming out of penis** | Yes | 597(99.7) |
| | No | 2 (0.3) |
| | Total | 599 (100) |
| **Nature of Urethral discharge** | Profuse/Thick | 547 (91.3) |
| | Watery | 44 (7.3) |
| | Other | 8 (1.3) |
| | Total | 599 (100) |

**Table 2. Comparison of gram stains and culture methods for the detection of gonococcus isolates from urethral discharge specimens.**

| GC confirmation method | Result | N (%) |
|---|---|---|
| Gram stain | Positive | 449 (75) |
| | Negative | 150 (25) |
| Total | | 599(100) |
| Culture | Positive | 415 (69) |
| | Negative | 184 (31) |
| Total | | 599 (100) |

was higher (75%) by gram stain (Table 2). This is not surprising as considerable proportion of the patients (20%) were on antibiotics when the specimens were collected which would affect culture results (Table 3). The prevalence of gonococcus in this study was relatively higher than other studies conducted in Ethiopia [12–14]. The difference may be due to the nature of the participants in the current study considering all of them were males and also showing clinical manifestation of the disease. The general notion is that naturally, males tend to be more symptomatic for gonococcal infection and hence can have increased level of healthcare seeking behavior which in turn makes them statistically overrepresented [15].

## Antimicrobial resistance profile

In sub-Saharan Africa, gonococcal treatment practice is based on a syndromic approach using a single dose fluoroquinolone treatment. It is hypothesized that resistance to fluoroquinolones is low in Africa, but there has been limited systematic data collection and analysis to verify this notion. A multicounty antimicrobial resistance study on gonococcal strains isolated in 2004–2006 indicated low rates of fluoroquinolone resistance with 0%, 1.3% and 4.0% in the Central African Republic, Cameroon, and Madagascar, respectively [16]. Similarly, a study conducted in Maputo and Mozambique in 2005 suggested that there was no resistance to fluoroquinolone by gonococcal isolates [17].

**Table 3. Medical treatment history of patients with urethral discharge from health centers in Addis Ababa, August 2013–August 2014.**

| Medication history | Response | N (%) |
|---|---|---|
| Taking medication | Yes | 123 (20.5) |
| | No | 476 (79.5) |
| | Total | 599 (100) |
| Know the type of medication | Yes | 107 (87) |
| | No | 16 (13) |
| | Total | 123 (100) |
| Ciprofloxacin | Yes | 91 (85) |
| | No | 16 (15) |
| | Total | 107 (100) |
| Doxycycline | Yes | 93 (86.9) |
| | No | 14 (13.1) |
| | Total | 107 (100) |
| Metronidazole | Yes | 12 (11.2) |
| | No | 95 (88.8) |
| | Total | 107 (100) |

In contrast, the findings from several other countries in sub-Saharan Africa suggested increasing levels of fluoroquinolone resistance in gonococcal isolates. According to a study done in South Africa in 2004, 7% of the gonococcal isolates from the Pretoria region, 8% from the Western Cape, and 17% from Johannesburg were found to be resistant to antibiotics from the class of fluoroquinolone. In addition, another study conducted in same country and the same study populations in 2007 indicated that 27% of the gonococcal isolates from Cape Town and 32% of isolates from Johannesburg were found to be resistant to ciprofloxacin [18]. This represents a 2.9 fold and 1.9 fold increases, respectively, within a 3-year time period. Similarly, a two-year prospective study carried out among STI patients from 2004 to 2006 in Johannesburg indicated an increase in ciprofloxacin resistance from 13% in the first year to 26.3% in the second year [19]. Another study conducted in Kenya and Uganda also showed that gonococcal resistance level to fluoroquinolone has reached up to 53% and 83%, respectively [20, 21].

The present study has revealed that *N. gonorrhoeae* isolates recovered from urethral discharge of male patients in Addis Ababa, Ethiopia have shown a high level of resistance to the commonly prescribed fluoroquinolone class of antibiotics in Ethiopia (60%). This finding is in agreement with other studies which reported resistance levels between 53% and 83% in the East African region including Kenya and Uganda. Reports from South Africa also indicate that the resistance level has reached up to 32% [19–22]. The proportion of ciprofloxacin-resistant gonococcal isolates in the United States has also reached more than 30% [3]. The high proportion of quinolone resistance in this study might be due to prior treatment using ciprofloxacin, as indicated in Table 3. Gonococcal syndromic treatment using oral fluoroquinolone has become very problematic due to the emergence of a high proportion of resistant isolates, as witnessed from the current study. The good news, however, is that non-susceptibility to ceftriaxone has not been detected in any of the isolates tested during the study period. This finding is not in agreement with other studies conducted in different part of Ethiopia [12–14]. This may be due to exposure of participants to a specific group of antimicrobial agents during the study period. However, our finding was in agreement with studies conducted elsewhere [23–27]. As indicated in Fig 1, majority of the isolates have shown MIC value of 0.016 μg/ml for ceftriaxone with all of them having MIC values well below the cut-off point (0.25 μg/ml). However, the existence of certain segments of the isolate population with MIC values close to the cut-off point may indicate potential for a minority non-susceptible bacterial population to potentially replace the susceptible majority population. Therefore, investigating the molecular mechanism of resistance in these group of isolates may be imperative to fully understand the epidemiology [26].

The figure shows the MIC levels of ceftriaxone for gonococcal isolates recovered from urethral discharge specimens that were collected from male patients. All of the isolates tested using concentration gradient strips (E-test) were well below the cut-off point (0.25 μg/ml) for ceftriaxone MIC, with majority of them having MIC value of 0.016 μg/ml and none of them being non-susceptible. A small proportion of the isolates had MIC value at the cut-off point.

In our study, the Penicillinase test was carried out by a chromogenic test showing almost more than half of the isolates to be positive for beta lactamase. Most of the isolates in the current study were resistant to Benzyl penicillin even though the antibiotic is not used for the national gonococcal treatment algorithm (Table 4). This finding from our study was also in line with other studies [24, 25, 27–29].

According to WHO, dual therapy is the preferred option for treatment of gonococcal infection instead of single therapy [1]. In the present study, non-susceptibility for the combination of ciprofloxacin and penicillin was observed at a rate of 6.9% (25/361), while for ciprofloxacin and spectinomycin were at a rate of 0.8% (3/361). Institutionalizing a surveillance system in the country might help track the resistance level of the isolates.

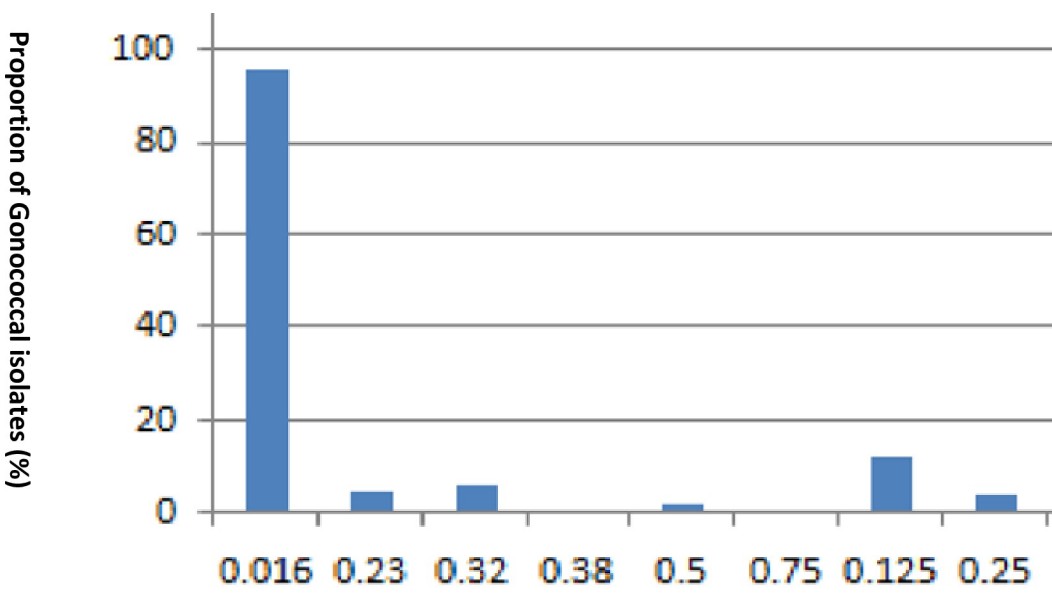

**Fig 1. Minimum Inhibitory Concentration (MIC) of ceftriaxone against *N. gonorrhoeae*.**

Formerly, the STI treatment practice in Ethiopia was based on the syndromic approach (treating individuals immediately for possible causes of STI syndromes based on symptom). The syndromic treatment guideline was produced in 2006 and has been in use for years [10]. However, because of the findings from this study and other studies in the country, the national guideline for treatment has been changed [30]. In the former guideline, ciprofloxacin was recommended to treat gonococcal infections because resistance to it was not then documented. The current guideline recommends ceftriaxone instead of ciprofloxacin [30]. The etiologic approach to diagnosis of gonorrhea is important, especially for revisiting clients, in order to identify non-susceptible isolates to serve as candidates for antimicrobial agents in practice.

## Conclusions

In the current study, the proportion of *N. gonorrhoeae* isolates in males with urethral discharge in Addis Ababa, Ethiopia, was found to be high. Of greatest concern was the finding that these

**Table 4. Percent of antimicrobial susceptibility pattern of gonococcal isolates recovered from urethral discharge of patients (N = 361).**

| Antimicrobial agent | Class | Susceptibility profile | | | |
|---|---|---|---|---|---|
| | | Resistant, n (%) | Intermediate, n (%) | Susceptible, n (%) | Non susceptible, n (%) |
| P | Penicillins | 191(52.9) | 0(0) | 170 (47.1) | - |
| Sp | Aminocyclitols | 11 (3) | 4 (1.1) | 346 (95.9) | - |
| Cip | Flouroquinolone | 217 (60.2) | 26 (7.1) | 118 (32. 7) | - |
| CRO | Cephalposrin | 0 (0) | 0(0) | 361 (100) | - |
| CFX | Cephalposrin | - | 0(0) | 307 (85) | 54 (15) |
| CTX | Cephalosporin | 4 (1.1) | 0(0) | 357 (98.9) | - |
| AZ | Macrolides | 36 (10) | 0(0) | 325 (90) | - |

AZ = Azithromycin, CIP = Ciprofloxacin, CFX = Cefixime, CRO = Ceftriaxone, CTX = Cefoxitin, P = Penicilin, SP = Spectinomycin.

gonococcal isolates were highly resistant to the new generation of antibiotics, fluoroquinolones (ciprofloxacin), which has been indicated as the treatment of choice according to previous national guidelines. Results generated from this study were used as input to revise the national syndromic guidelines for management of patients presenting with urethral discharge due to gonorrhea. As a result, ciprofloxacin was replaced by ceftriaxone which was found to be effective in terms of in vitro susceptibility results [30]. In conclusion, the syndromic-based diagnostic approach needs to be periodically validated and modified based on determination of susceptibility patterns of *N. gonorrhoeae* isolates in the region. Since this study was done in 2014, additional studies are warranted to understand the current antimicrobial resistance status of *gonorrhoeae* isolates in Ethiopia.

## Supporting information

**S1 Data.**
(XLSX)

## Acknowledgments

We are grateful to the Ethiopian Public Health Institute for permitting us to conduct this study. We would also like to thank Dr. Laura Binkley, from Global One Health initiative- the Ohio State University, for editing the manuscript.

## Author Contributions

**Conceptualization:** Surafel Fentaw.

**Formal analysis:** Eyasu Tigabu.

**Methodology:** Surafel Fentaw, Negga Asamene, Meseret Assefa.

**Writing – original draft:** Surafel Fentaw, Rajiha Abubeker.

**Writing – review & editing:** Yonas Bekele, Eyasu Tigabu.

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
