## [Decision Letter · Decision Letter 0]

31 Dec 2019

PONE-D-19-32111

Antimicrobial Susceptibility Profile of Gonococcal Isolates Obtained from Men Presenting with Urethral Discharge: Implication for National Syndromic Treatment guideline

PLOS ONE

Dear Mr Dinku,

Thank you for submitting your manuscript to PLOS ONE. After careful consideration, we feel that it has merit but does not fully meet PLOS ONE’s publication criteria as it currently stands. Therefore, we invite you to submit a revised version of the manuscript that addresses the points raised during the review process.

We would appreciate receiving your revised manuscript by Feb 14 2020 11:59PM. To enhance the reproducibility of your results, we recommend that if applicable you deposit your laboratory protocols in protocols.io, where a protocol can be assigned its own identifier (DOI) such that it can be cited independently in the future. For instructions see: http://journals.plos.org/plosone/s/submission-guidelines#loc-laboratory-protocols

We look forward to receiving your revised manuscript.

Kind regards,

David Gabriel Regan, Ph.D.

Academic Editor

PLOS ONE

Journal Requirements:

2. Please ensure that you include a title page within your main document. You should list all authors and all affiliations as per our author instructions and clearly indicate the corresponding author.

Additional Editor Comments (if provided):

Please revise this manuscript addressing in full all the comments and suggestions the reviewers have provided. In particular pay attention to providing all the data, and ensuring that all figures and tables are properly presented, annotated and referenced and discussed in the text. It is recommended that you seek editorial assistance to ensure the manuscript is written in grammatically correct, intelligible, standard English.

Reviewers' comments:

Reviewer's Responses to Questions

**Comments to the Author**

1. Is the manuscript technically sound, and do the data support the conclusions?

Reviewer #1: Yes

Reviewer #2: Partly

2. Has the statistical analysis been performed appropriately and rigorously? 

Reviewer #1: Yes

Reviewer #2: Yes

3. Have the authors made all data underlying the findings in their manuscript fully available?

Reviewer #1: No

Reviewer #2: No

4. Is the manuscript presented in an intelligible fashion and written in standard English?

Reviewer #1: No

Reviewer #2: No

5. Review Comments to the Author

Reviewer #1: The manuscript describes a study of urethral discharges from male patients in Addis Ababa, Ethiopia. It is motivated by the growing problem of antibiotic resistance in Neisseria Gonorrhoea. Its main findings are that gonococcal isolates are detected in 69% of those discharges. Of these isolates, a significant proportion were non-susceptible to fluoroquinolone, which is part of the basis of gonorrhoea management in sub-Saharan African countries including Ethiopia. The scientific work is sound but considerable improvement is required in its presentation.

1) The manuscript appears to be technically sound. The study is described with an appropriate level of detail and its design was clearly suitable for the topic of interest so that one would expect its data to be capable of supporting its conclusions. However the actual data are not supplied.

2) The statistical analysis is appropriate to the question at hand and done correctly.

3) The authors have not made their data available. It is not in the manuscript, nor is there a link provided, and there is no supplementary section.

4) The English in this manuscript does not meet the standards expected for a credible, reviewed scientific journal. It is intelligible most of the time but replete with grammatical errors and poorly worded phrases, too many to outline here. An exception is paragraph five in "Results and Discussion" in which figure one is referenced and discussed. The authors presumably mean to say that ceftriaxone non-susceptibility was not found in any isolates. The rest of this paragraph is incomprehensible. The authors are advised to rewrite the manuscript with assistance from a native speaker of English.

Additional Comments:

A) The presentation of percentages throughout the manuscript is inconsistent, most notably in the third paragraph of the "Results and Discussion" section and in table 1. It is indicated that percentages are being presented but some figures have "%" signs while others do not. The "%" should be used in text while the convention used in table 3 should be used consistently in all tables.

b) Table 2 is not referenced in the text although it is apparently relevant to the fourth paragraph in "Results and Discussion". Also, most of its data appears to be missing which renders the table confusing. As far as I can make out, each row should add up to 100 (%) but they do not. This table should be either corrected and properly referenced, or deleted.

c) The page with table 2 has what appears to be a footnote entitled "MIC Ranges for Resistance and Intermediate Susceptibility as defined by CLSI Guidelines". This useful and relevant information should be clearly presented in its own table and appropriately referenced from within the text.

d) The bars in table 1 have no apparent order. They should be plotted in order and positioned to scale on the x-axis since these categories are defined by the quantifiably measurable MIC.

e) It is stated in the final paragraph of the "Results and Discussion" section and again in the "Conclusions" that this work has already led to a revision of Ethiopia's national guidelines for managing patients with urethral discharge, but no description or reference to a description is given. The revisions should be described and ideally a paper or relevant government document should be referenced.

f) References to tables and figures should use a capital letter, "Table 3" instead of "table 3". This was sometimes neglected, particularly for Table 3 and Figure 1.

Reviewer #2: Antimicrobial-resistant gonorrhea is a major public health threat. Therefore, manuscript addressing the antimicrobial susceptibility of Neisseria gonorrhoeae (NG) isolates are important. However, this manuscript, as written, lacks the necessary level of detail and scientific rigor to be published in PLOS one Medicine.

Major issues.

Introduction.

1) Does not describe the current WHO recommended treatment for gonorrhea. This is important in order to provide context for the recommended treatment in Ethiopia.

2) There are several recent papers describing the susceptibility of NG isolates in Africa which have found that most NG isolates from African countries are resistant to Ciprofloxacin.

3) The references cited are older. A quick search on Pubmed could yield a list of recent publications on this topic.

4) Overall, the introduction is too short.

Results/discussion

1) Although table 1 describes the gram stain results, a discussion of gram stain results in comparison to culture results should be included.

2) The authors need to be careful in attributing the high level of NG infections to the fact that the samples were collected from men. The reason for the high positivity rates is because the samples were collected from symptomatic subjects.

3) The discussion on the implication for National syndromic treatment guideline need to be expanded. It is currently limited to one line. “Antimicrobial Susceptibility Profile of Gonococcal Isolates Obtained from Men Presenting with Urethral Discharge: Implication for National Syndromic Treatment guideline”

The title should also include Ethiopia to better guide the reader.

Overall, this manuscript is suitable for a journal like STD which is more related to public health

6. PLOS authors have the option to publish the peer review history of their article (what does this mean?). If published, this will include your full peer review and any attached files.

Reviewer #1: No

Reviewer #2: No

---

## [Author Response · Author response to Decision Letter 0]

28 Feb 2020

Dear Editor 

Thank you for your constructive comments. We have tried to address all the comments accordingly

---

## [Decision Letter · Decision Letter 1]

24 Mar 2020

PONE-D-19-32111R1

Antimicrobial susceptibility profile of Gonococcal isolates obtained from men presenting with urethral discharge in Addis Ababa, Ethiopia: Implication for national syndromic treatment guideline

PLOS ONE

Dear Mr Dinku,

Thank you for submitting your manuscript to PLOS ONE. After careful consideration, we feel that it has merit but does not fully meet PLOS ONE’s publication criteria as it currently stands. Therefore, we invite you to submit a revised version of the manuscript that addresses the points raised during the review process.

We would appreciate receiving your revised manuscript by May 08 2020 11:59PM. To enhance the reproducibility of your results, we recommend that if applicable you deposit your laboratory protocols in protocols.io, where a protocol can be assigned its own identifier (DOI) such that it can be cited independently in the future. For instructions see: http://journals.plos.org/plosone/s/submission-guidelines#loc-laboratory-protocols

We look forward to receiving your revised manuscript.

Kind regards,

David Gabriel Regan, Ph.D.

Academic Editor

PLOS ONE

Additional Editor Comments (if provided):

The reviewers agree that the revised version of the manuscript is is substantially improved but the language and grammatical issues have not been adequately addressed. It was recommended that the authors seek editorial assistance in addressing these issues and it seems this recommendation has not been taken on board. The reviewers agree that the manuscript is scientifically sound on the whole but the language and presentation is not up to the standard required for publication in PLoS ONE. I feel the authors should have a final opportunity to address the language and presentation issues highlighted by the reviewers but the manuscript will need to be substantially improved for me to be able to recommend publication in PLoS ONE. The authors also need to provide a consistent statement that aligns with PLoS ONE publication policy in regard to availability of data.

Reviewers' comments:

Reviewer's Responses to Questions

**Comments to the Author**

1. If the authors have adequately addressed your comments raised in a previous round of review and you feel that this manuscript is now acceptable for publication, you may indicate that here to bypass the “Comments to the Author” section, enter your conflict of interest statement in the “Confidential to Editor” section, and submit your "Accept" recommendation.

Reviewer #1: (No Response)

Reviewer #2: (No Response)

2. Is the manuscript technically sound, and do the data support the conclusions?

Reviewer #1: Yes

Reviewer #2: Yes

3. Has the statistical analysis been performed appropriately and rigorously? 

Reviewer #1: Yes

Reviewer #2: Yes

4. Have the authors made all data underlying the findings in their manuscript fully available?

Reviewer #1: No

Reviewer #2: Yes

5. Is the manuscript presented in an intelligible fashion and written in standard English?

Reviewer #1: No

Reviewer #2: No

6. Review Comments to the Author

Reviewer #1: The manuscript has been considerably improved in both the discussion of content and in the quality of its writing. Unfortunately it still falls well short of the standard expected for a credible scientific journal. There is still a large number of spelling and basic grammatical errors, clearly indicating that a native speaker was not consulted as advised. Other sentences were either badly worded, or too long and needing to be broken into smaller sentences. Examples are the final sentence of the abstract and "Globally, more than millions of curable STD are affecting people every day" from the Background section.

The caption for Figure 1 inexplicably appears in isolation in the middle of page nine with no figure. This may be an artifact of separating figures and tables from the main body of the text in the draft copy but the same has not happened for the tables so care must be taken with this.

Table 3 (previously Table 2) is now much clearer but the percentages in the bottom two rows still do not add to 100%. My best guess is that the remaining samples were non-susceptible, but this should be indicated explicitly.

Tables 3 and 4 are referenced out-of-order and their numbering should be reversed.

The y-axis of the figure is labelled "Proportion of Gonococcal isolates" but the bars clearly do not add up to 100%. Either it is mislabelled or the scale is wrong. If the axis is a proportion then the ticks should be labelled 20%, 40%, ... , 100% .

The authors are inconsistent in their responses to PLOS ONE's data sharing requirement. They have claimed under their Data Availability statement that "data are fully available without restriction" and "included in the manuscript" and that they "can give the raw data", but stated in their response to reviewers that they are restricted from doing so from privacy concerns. I suspect that they do not properly understand this requirement. The PLOS Data policy allows exceptions if the data "compromise the privacy or confidentiality of human research subjects" but the de-identification of data in a study such as this has long been standard practice and there is no discussion of these restrictions in the Materials and Methods.

While the scientific core of this work appears to be sound, the manuscript and data-sharing still do not approach the standards expected for a high quality scientific journal. I suggest after attending to these comments that the authors submit their work to a journal more specific to public health.

Reviewer #2: Thanks to the authors for making the required changes.

Please consider making the following revisions.

1) Abstract. Please note that gonorrhea is the disease/infection. No need to see gonorrhea disease.

2) Introduction. Please consider editing the following sentence "Globally, more than millions of curable STD are

affecting people every day. According to WHO, in 2012 alone, there were an estimated 78 million new

case of gonococcal diseases [1].

3) Introduction: The following section of the introduction could use some editing. "Over the past few years, gonococcus have

become less susceptible to previously used antibiotics such as sulfonamides or tetracycline. Until recently, quinolones and third-generation cephalosporins are alternative to treat gonococcal infections". Please note that sulfonamides have not been recently prescribed and that third generation celaphosporins are currently the recommended treatment option.

4) Methods. STI needs to be spelled out the first time that is used in the document. Overall, this reviewer recommends changing STD to STI.

5) Methods. The following two sentences are confusing. "A sterile cotton-tipped swab were used to obtain a swab specimens. Then sterile Dacron swabs tipped applicator were used to collect urethral secretions". It sounds like two different swabs were used for sample collection.

6) Operating definitions. Penicillin is mentioned here, but in the methods section it is not clear that the isolates were tested for penicillin susceptibility.

7) Ehtics and Consent. Please clarify the following sentence. "At the enrollment visit, all men with urethral discharge (UD) were given written consent diagnosed according to the syndromic treatment guidelines approved in Ethiopia". It is not clear to this reviewer what "written consent diagnosed" means.

8) Results. This reviewer does not understand the relevance of the following sentence. "Observation of the specimens revealed that over 90% of them were profuse /thick discharge (Table-1).

9) Proportion of gonococcal isolates recovered. The sentence "Compared to culture, the proportion of the identified isolates was higher (75%) when gram stain method was used (Table-2)" requires editing. It is not possible to isolate NG by gram stain. Please edit sentence as follows "Compared to culture, the proportion of presumptive gonorrhea-positive samples was higher (75%) by gram stain.

10) Figure 1 header. Please change N. gonorrhea to N. gonorrhoeae.

11) Conclusion. Please change gonorrhea disease to gonorrhea.

12) Conclusion. The authors should add a sentence regarding the limitation of using samples that were collected in 2013-2014. The epidemiology of resistance is probably different now. They can also say that additional studies are warranted to understand the current (2020) epidemiology of antimicrobial resistance in NG in Ethiopia.

7. PLOS authors have the option to publish the peer review history of their article (what does this mean?). If published, this will include your full peer review and any attached files.

Reviewer #1: Yes: Michael Luke Walker

Reviewer #2: No

---

## [Author Response · Author response to Decision Letter 1]

2 May 2020

2. Is the manuscript technically sound, and do the data support the conclusions?

Reviewer #1: Yes

Reviewer #2: Yes

Response to reviewers:

Dear reviewers, thank you so much.

3. Has the statistical analysis been performed appropriately and rigorously? 

Reviewer #1: Yes

Reviewer #2: Yes

Response to reviewers:

Dear reviewers, thank you so much. 

4. Have the authors made all data underlying the findings in their manuscript fully available?

Reviewer #1: No

Reviewer #2: Yes

Response to reviewers:

Dear reviewers, thank you so much. As to the availability of Data, we have requested our institute for de-identifying the data set and are able to share it, as supplement file, during this current submission. Thank you. 

5. Is the manuscript presented in an intelligible fashion and written in standard English?

Reviewer #1: No

Reviewer #2: No

Response to reviewers:

Dear reviewers, thank you. We have tried to address the major issue raised accordingly. We sought an editorial assistance from native English speaker, Dr. Laura Binkley (binkley.69@osu.edu) from the Ohio State Unviersity, USA. And we thank Dr. Laura for the English language edits and overall comments that, we believe, has substantially improved the manuscript. We have checked the spelling and over all neatness of the manuscript in as much as we can. 

6. Review Comments to the Author

Reviewer #1: The manuscript has been considerably improved in both the discussion of content and in the quality of its writing. Unfortunately it still falls well short of the standard expected for a credible scientific journal. There is still a large number of spelling and basic grammatical errors, clearly indicating that a native speaker was not consulted as advised. Other sentences were either badly worded, or too long and needing to be broken into smaller sentences. Examples are the final sentence of the abstract and "Globally, more than millions of curable STD are affecting people every day" from the Background section.

Response to reviewer: 

Dear reviewer, thank you for the priceless comments. We have tried to address your comments and have consulted a native English speaker to address the language issues. Thank you once again. 

The caption for Figure 1 inexplicably appears in isolation in the middle of page nine with no figure. This may be an artifact of separating figures and tables from the main body of the text in the draft copy but the same has not happened for the tables so care must be taken with this.

Response to reviewer: 

Dear reviewer, thank you for the comment; as you rightly put it, it was an artifact left during the process of separating the figure from the main body of the text. We are very sorry for killing your precious time. We have made the correction. We have also tried to check to avoid such mistakes all over the manuscript. Thank you. 

Table 3 (previously Table 2) is now much clearer but the percentages in the bottom two rows still do not add to 100%. My best guess is that the remaining samples were non-susceptible, but this should be indicated explicitly. Tables 3 and 4 are referenced out-of-order and their numbering should be reversed.

Response to reviewer: 

Dear reviewer, thank you for the comments and positive words. The numbers were simply counted for those sample how have CIP + Spectinomycin combined resistance level. To avoid confusion we have omitted the bottom two rows that do not add up to 100. We have also revised the position of table-3 and 4. Thank you. 

The y-axis of the figure is labelled "Proportion of Gonococcal isolates" but the bars clearly do not add up to 100%. Either it is mislabelled or the scale is wrong. If the axis is a proportion then the ticks should be labelled 20%, 40%, ... , 100% .

Response to reviewer: 

Thank you. Y-axis is proportion but we did not indicate in percentage. We have corrected this by adding % on the y-axis label. 

The authors are inconsistent in their responses to PLOS ONE's data sharing requirement. They have claimed under their Data Availability statement that "data are fully available without restriction" and "included in the manuscript" and that they "can give the raw data", but stated in their response to reviewers that they are restricted from doing so from privacy concerns. I suspect that they do not properly understand this requirement. The PLOS Data policy allows exceptions if the data "compromise the privacy or confidentiality of human research subjects" but the de-identification of data in a study such as this has long been standard practice and there is no discussion of these restrictions in the Materials and Methods.

Response to reviewer: 

Dear Reviewer, You are right we did not fully understood the policy but after the comments, this issue has been addressed by provision of the de-identified data set as a supplement file with this submission. 

While the scientific core of this work appears to be sound, the manuscript and data-sharing still do not approach the standards expected for a high quality scientific journal. I suggest after attending to these comments that the authors submit their work to a journal more specific to public health.

Response to reviewers: 

Dear reviewer, thank you so much for the very constructive comments you gave us and definitely that have significantly improved our manuscript in terms of language usage and data presentation. Thank you for your precious time and patience in reviewing our manuscript. 

Reviewer #2: Thanks to the authors for making the required changes.

Please consider making the following revisions.

1) Abstract. Please note that gonorrhea is the disease/infection. No need to see gonorrhea disease.

Response to reviewers: 

Dear reviewer, thank you so much for the very constructive comments. The correction have been made as per the comment. 

2) Introduction. Please consider editing the following sentence "Globally, more than millions of curable STD are

affecting people every day. According to WHO, in 2012 alone, there were an estimated 78 million new

case of gonococcal diseases [1].

Response to reviewers: 

Dear reviewer, thank you so much for the comments. The correction have been made as per the comment. 

3) Introduction: The following section of the introduction could use some editing. "Over the past few years, gonococcus have

become less susceptible to previously used antibiotics such as sulfonamides or tetracycline. Until recently, quinolones and third-generation cephalosporins are alternative to treat gonococcal infections". Please note that sulfonamides have not been recently prescribed and that third generation celaphosporins are currently the recommended treatment option.

Response to reviewers: 

Dear reviewer, thank you so much for the comments. The correction have been made as per the comment. 

4) Methods. STI needs to be spelled out the first time that is used in the document. Overall, this reviewer recommends changing STD to STI.

Response to reviewers: 

Dear reviewer, thank you so much for the comments. The correction have been made as per the comment. And STD was replaced with STI.

5) Methods. The following two sentences are confusing. "A sterile cotton-tipped swab were used to obtain a swab specimens. Then sterile Dacron swabs tipped applicator were used to collect urethral secretions". It sounds like two different swabs were used for sample collection.

Response to reviewers: 

Dear reviewer, thank you so much for the comment. It was Dacron swab that was used. Correction has been made accordingly. Thank you once again. 

6) Operating definitions. Penicillin is mentioned here, but in the methods section it is not clear that the isolates were tested for penicillin susceptibility.

Response to reviewers: 

Dear reviewer, thank you so much for the comments. Penicillin was unintentionally omitted. The correction have been made as per the comment. 

7) Ehtics and Consent. Please clarify the following sentence. "At the enrollment visit, all men with urethral discharge (UD) were given written consent diagnosed according to the syndromic treatment guidelines approved in Ethiopia". It is not clear to this reviewer what "written consent diagnosed" means.

Response to reviewers: 

Dear reviewer, thank you so much for the comments. The sentence has been clarified per the comment given. 

8) Results. This reviewer does not understand the relevance of the following sentence. "Observation of the specimens revealed that over 90% of them were profuse /thick discharge (Table-1).

Response to reviewers: 

Dear reviewer, thank you so much for the comments. All the urethral discharge specimens were observed visually for checking the clinical manifestation. 

9) Proportion of gonococcal isolates recovered. The sentence "Compared to culture, the proportion of the identified isolates was higher (75%) when gram stain method was used (Table-2)" requires editing. It is not possible to isolate NG by gram stain. Please edit sentence as follows "Compared to culture, the proportion of presumptive gonorrhea-positive samples was higher (75%) by gram stain.

Response to reviewers: 

Dear reviewer, thank you so much for the comments. The sentence was not meant to mean that we isolated using gram stain. Since gram stain, in resource limiting setting like ours, can be using to guide the treatment of patients (male), we felt to compare both. We have amended the sentence as commented. Thanks you for your helpful comment. 

10) Figure 1 header. Please change N. gonorrhea to N. gonorrhoeae.

Response to reviewers: 

Dear reviewer, thank you so much for the comments. Corrections have been made per the comment. 

11) Conclusion. Please change gonorrhea disease to gonorrhea.

Response to reviewers: 

Dear reviewer, thank you so much for the comments. Corrections have been made per the comment. 

12) Conclusion. The authors should add a sentence regarding the limitation of using samples that were collected in 2013-2014. The epidemiology of resistance is probably different now. They can also say that additional studies are warranted to understand the current (2020) epidemiology of antimicrobial resistance in NG in Ethiopia.

Response to reviewers: 

Dear reviewer, thank you so much for the comments. The issue has been addressed as per the given comment.

---

## [Editor Report · Decision Letter 2]

7 May 2020

PONE-D-19-32111R2

Antimicrobial susceptibility profile of Gonococcal isolates obtained from men presenting with urethral discharge in Addis Ababa, Ethiopia: Implications for national syndromic treatment guideline

PLOS ONE

Dear Mr Dinku,

Thank you for submitting your manuscript to PLOS ONE. After careful consideration, we feel that it has merit but does not fully meet PLOS ONE’s publication criteria as it currently stands. Therefore, we invite you to submit a revised version of the manuscript that addresses the points raised during the review process.

We would appreciate receiving your revised manuscript by Jun 21 2020 11:59PM. To enhance the reproducibility of your results, we recommend that if applicable you deposit your laboratory protocols in protocols.io, where a protocol can be assigned its own identifier (DOI) such that it can be cited independently in the future. For instructions see: http://journals.plos.org/plosone/s/submission-guidelines#loc-laboratory-protocols

We look forward to receiving your revised manuscript.

Kind regards,

David Gabriel Regan, Ph.D.

Academic Editor

PLOS ONE

Additional Editor Comments (if provided):

The authors have, on the whole, addressed the concerns expressed by the reviewers and the manuscript is much improved. There remain a couple of minor issues that need to be addressed before I can recommend this manuscript for publication;

1) the frequency columns in tables 1-3 are redundant and should be removed because the identical information in provided in the 'N (%)' column

2) the total for Gram stain is missing in Table 2

3) Table 4 does not contain information regarding multi-drug non-susceptibility as stated on page 9. This information needs to be added to the table or presented in a separate table.

4) The results presented in Figure 1 are not adequately described in the results on page 9 and the cut-off MIC is not specified so this can not be interpreted. There is also no figure legend for Figure 1, only a figure title.

---

## [Author Response · Author response to Decision Letter 2]

11 May 2020

The authors have, on the whole, addressed the concerns expressed by the reviewers and the manuscript is much improved. There remain a couple of minor issues that need to be addressed before I can recommend this manuscript for publication;

Response to the Editor:

Dear editor, thank you so much for all the constructive comments. And thank you also for this last opportunity, again, for us to be able to publish our finding. We have tried to address the major issue raised accordingly.

1) the frequency columns in tables 1-3 are redundant and should be removed because the identical information in provided in the 'N (%)' column

Response to the Editor:

The correction has been made. The frequency columns in all tables has been removed. Thank you so much for due diligence. 

2) the total for Gram stain is missing in Table 2

Response to the Editor

The correction has been made. The total for gram stain has been added. Thank you so much for due diligence.

3) Table 4 does not contain information regarding multi-drug non-susceptibility as stated on page 9. This information needs to be added to the table or presented in a separate table.

Response to the Editor:

Correction has been made to the proportion of isolates with non-susceptibility to ciprofloxacin and pencillin and ciprofloxacilin and spectinomycin (In combination) and the reference made to table-4 was removed because it is presented in a narrative form in page 9. In the previous submissions, we added the information in the bottom 2 rows of table -4 and later omitted it because that has caused a confusion to the reviewers as the numbers in these rows did not add up to hundred. Since we only had the combined non-susceptibility for the two agents so did not feel to put the information in a table of its own rather we have indicated in the narrative on page 9. 

4) The results presented in Figure 1 are not adequately described in the results on page 9 and the cut-off MIC is not specified so this cannot be interpreted. There is also no figure legend for Figure 1, only a figure title.

 Response to the Editor:

Thank you for the comment. We have attempted to describe the result presented in the figure on page 9 and also have included figure legend as commented on page 9 right immediately after the paragraph where the figure was referenced.

---

## [Editor Report · Decision Letter 3]

13 May 2020

Antimicrobial susceptibility profile of Gonococcal isolates obtained from men presenting with urethral discharge in Addis Ababa, Ethiopia: Implications for national syndromic treatment guideline

PONE-D-19-32111R3

Dear Dr. Dinku,

We are pleased to inform you that your manuscript has been judged scientifically suitable for publication and will be formally accepted for publication once it complies with all outstanding technical requirements.

With kind regards,

David Gabriel Regan, Ph.D.

Academic Editor

PLOS ONE

Additional Editor Comments (optional):

I commend the authors on this work and for addressing all issues raised by me and the reviewers.
---

## [Editor Report · Acceptance letter]

20 May 2020

PONE-D-19-32111R3 

Antimicrobial susceptibility profile of Gonococcal isolates obtained from men presenting with urethral discharge in Addis Ababa, Ethiopia: Implications for national syndromic treatment guideline 

Dear Dr. Fentaw:

I am pleased to inform you that your manuscript has been deemed suitable for publication in PLOS ONE. Congratulations! Your manuscript is now with our production department. 

With kind regards,

on behalf of

Associate Professor David Gabriel Regan 

Academic Editor

PLOS ONE